# Computing Localized Breakthrough Curves and Velocities of Saline Tracer from Ground Penetrating Radar Monitoring Experiments in Fractured Rock

**Peter-Lasse Giertzuch** [1,*], **Alexis Shakas** [1], **Joseph Doetsch** [1,2], **Bernard Brixel** [3], **Mohammadreza Jalali** [4] **and Hansruedi Maurer** [1]

1    Institute of Geophysics, ETH Zurich, 8092 Zurich, Switzerland; alexis.shakas@erdw.ethz.ch (A.S.); joseph.doetsch@alumni.ethz.ch (J.D.); hansruedi.maurer@erdw.ethz.ch (H.M.)
2    Lufthansa Industry Solutions, 65479 Raunheim, Germany
3    Geological Institute, ETH Zurich, 8092 Zurich, Switzerland; bernard.brixel@erdw.ethz.ch
4    Chair of Engineering Geology and Hydrogeology, RWTH Aachen, 52056 Aachen, Germany; jalali@lih.rwth-aachen.de
*    Correspondence: peter-lasse.giertzuch@erdw.ethz.ch

**Abstract:** Solute tracer tests are an established method for the characterization of flow and transport processes in fractured rock. Such tests are often monitored with borehole sensors which offer high temporal sampling and signal to noise ratio, but only limited spatial deployment possibilities. Ground penetrating radar (GPR) is sensitive to electromagnetic properties, and can thus be used to monitor the transport behavior of electrically conductive tracers. Since GPR waves can sample large volumes that are practically inaccessible by traditional borehole sensors, they are expected to increase the spatial resolution of tracer experiments. In this manuscript, we describe two approaches to infer quantitative hydrological data from time-lapse borehole reflection GPR experiments with saline tracers in fractured rock. An important prerequisite of our method includes the generation of GPR data difference images. We show how the calculation of difference radar breakthrough curves (DRBTC) allows to retrieve relative electrical conductivity breakthrough curves for theoretically arbitrary locations in the subsurface. For sufficiently small fracture apertures we found the relation between the DRBTC values and the electrical conductivity in the fracture to be quasi-linear. Additionally, we describe a flow path reconstruction procedure that allows computing approximate flow path distances using reflection GPR data from at least two boreholes. From the temporal information during the time-lapse GPR surveys, we are finally able to calculate flow-path averaged tracer velocities. Our new methods were applied to a field data set that was acquired at the Grimsel Test Site in Switzerland. DRBTCs were successfully calculated for previously inaccessible locations in the experimental rock volume and the flow path averaged velocity field was found to be in good accordance with previous studies at the Grimsel Test Site.

**Keywords:** GPR; tracer test; fractured rock; flow and transport; reservoir monitoring

## 1. Introduction

Tracer experiments are one of the standard tools to characterize flow and transport processes in fractured rock and to assess flow geometries and hydraulic connectivity. Tracers are typically injected into open, permeable fractures, and their transport is monitored by measuring their concentration at discrete observation locations with appropriate sensors (e.g., [1,2]). There exists a large variety of tracers, such as saline tracers, (fluorescent) dyes, radionuclides, and DNA labels, which have been successfully applied for the characterization of flow paths related with geothermal reservoirs (e.g., [3–5]). The temporal variation of the tracer concentrations at the observation locations can be characterized using breakthrough curves (BTC). Besides determining hydraulic connections, such curves

facilitate analyses of the flow systems through statistical properties, for example tracer swept volumes, fluid residence time distributions, and tracer velocities (e.g., [6,7]). Such information is critical to a large number of subsurface applications, ranging from groundwater remediation to geothermal heat extraction. Transport processes in fractured rock are typically characterized by non-Fickian behavior, which can arise from variations in fracture permeability, flow path connections, and fracture-matrix interactions [8]. However, solute tracer tests typically only provide spatially sparse observations and therefore flow and transport properties have to be interpolated or upscaled between observation locations. Numerical simulations are used to investigate the relevant processes (e.g., [9]), but can only be sparsely constrained or appraised through field data.

Besides solute tracers, thermal tracers have also been used more frequently over the last years, due to the relevance of heat transport for geothermal reservoirs (e.g., [10]). Heat tracers have also been combined with solute tracers to investigate thermal attenuation and lag time in fractured rock [11]. To monitor heat transport in such experiments, fiber-optic distributed temperature sensing can be used, which allows to resolve thermal breakthroughs spatially along the deployment boreholes. However, the monitoring capabilities for fluid flow and tracer transport are still strongly dependent on sensor deployment possibilities and the hydraulic connections to the discrete monitoring locations. Equally limited by reservoir access locations are hydraulic tests that can be used for reservoir characterizations. Besides determining hydraulic parameters such as transmissivities, permeabilities and hydraulic conductivities (e.g., [12,13]), such tests also allow for investigations on flow paths and their connectivity (e.g., [14]), they are dependent on discrete observation locations. Therefore, for the characterization of flow and transport processes in fractured rock, a method that can remotely retrieve information on tracer transport with high spatial resolution at various locations in the subsurface without the need of direct access points would be desirable.

Conceptually, it is possible to employ repeated ground penetrating radar (GPR) measurements for sensing spatial and temporal variations of appropriate tracers in saturated hydraulic systems. GPR makes use of electromagnetic waves in the frequency range of MHz to GHz. The propagation of the waves in the subsurface is primarily dependent on the dielectric permittivity $\varepsilon$ (often expressed as $\varepsilon_r = \varepsilon/\varepsilon_0$ with $\varepsilon_0$ being the vacuum permittivity) and the electrical conductivity $\sigma$ of the host medium, where $\varepsilon$ controls mainly the propagation velocity and $\sigma$ primarily determines the wave attenuation. Depending on these subsurface parameters and the used frequencies, GPR can be used to explore the subsurface in the range of centimeters to hundreds of meters. GPR systems typically consist of a transmitting and a receiving antenna and emit a short pulse that is reflected and/or transmitted at interfaces of contrasting electromagnetic properties (e.g., [15]). Besides surface-based GPR systems, also borehole GPR antennas exist that allow deployment at large depths (depending on the manufacturer between several 100 m to even the km range). For crystalline rock, the wave attenuation is low and thus large signal penetration depths can be achieved, which makes GPR also an option for deep reservoir characterization. To employ GPR as a tool for the characterization of tracer flow and transport, the tracer must have an electrical contrast (electrical conductivity or dielectric permittivity) to the fluids in the medium of interest. Electrically conductive tracers are an obvious choice. In contrast to classical sensor data, GPR measurements do not usually allow a fine temporal sampling, but, at least in principle, an almost continuous spatial sampling can be achieved.

Due to the possibility of enhanced spatial sampling, the use of GPR for hydrological research has been under investigation for some decades [16,17], but so far, there is no wide-spread application of this technology. In cross-hole transmission experiments, Day-Lewis et al. [18] were able to reconstruct the temporal propagation of a saline tracer plume through a 4D attenuation inversion scheme. Reflection GPR for fractured rock with small apertures was used by Talley et al. [19], who integrated amplitudes of reference and monitoring surface GPR data to create 2D saline tracer propagation maps. Theoretical calculations and confirmation experiments for conductivity-enhancing tracers have been

performed by Tsoflias and Becker [20], who found that lower frequencies appear to be more favorable for conductivity-change based GPR tracers, and they describe how changing conductivity will not only affect the reflection strength of a thin-bed, but also introduce a change in signal phase. In confirmation experiments, they could relate rising amplitudes to conductivity changes, by analyzing a single reflection peak in the GPR signal. Experiments showing an increase in relative GPR amplitude with rising conductivity levels due to tracer propagation in the subsurface were also successfully performed by Becker and Tsoflias [21]. By calibrating the measured GPR amplitudes through different steady-state conductivity experiments, they were able to quantify their measured monitoring results and translate GPR amplitude into conductivity. Dorn et al. [22] were able to apply GPR for saline tracer monitoring in fractured rock with borehole GPR antennas. Instead of analyzing amplitudes of monitoring GPR data, they applied a difference reflection imaging scheme, hence subtracting reference data from the monitoring data. Subtle changes in the GPR signal could be detected and they successfully related reflection changes of difference profiles (between reference and monitoring measurements) to conductivity changes within the fractures. Allroggen and Tronicke [23] demonstrated the use of time-lapse surface GPR during a soil irrigation experiment and calculated difference attributes over GPR trace windows, which they found to be more stable than data differences. Applying such attributes, they could visualize the temporal and spatial development of lateral preferential flow in the subsurface [24]. Hawkins et al. [25] used cross-polarized surface GPR to relate signal phase changes to tracer propagation and hence confirmed the phase change described in Tsoflias and Becker [20] to function as a tracer detection method. Shakas et al. [26] used time-lapse difference reflection imaging with borehole GPR to monitor saline tracer migration in push-pull experiments and were able to calculate GPR breakthrough curves by analyzing trace differences for different antenna positions. Furthermore, they successfully inferred apparent fracture apertures from these experiments [27]. Shakas et al. [28] could invert for the flow path of tracer on the fracture scale, revealing strong channeling effects during the tracer propagation. As saline tracers are denser than fresh-water, the transport behavior will be influenced by density effects, which was shown and overcome in Shakas et al. [29], by compensating the density difference with ethanol in push-pull tracer tests.

Lately, Giertzuch et al. [30] presented a data processing scheme for reflection borehole GPR data that allowed for tracer signal visualization in a challenging environment composed of many reflectors and fractures with sub-mm apertures. Furthermore, Giertzuch et al. [31] successfully combined two reflection borehole GPR data sets and data from a simultaneous transmission survey, for a 4D tracer flow reconstruction.

Here, we further push this concept of difference reflection imaging by constructing breakthrough curves from GPR data. Such an approach is attractive, since it allows the computation of breakthrough curves at locations that are inaccessible with traditional sensors. Furthermore, we exploit the information offered by the time-lapse GPR data to reconstruct flow path averaged tracer velocities as a 3D field. Opportunities and limitations of these two methodological developments are demonstrated with data from tracer experiments, conducted at the Grimsel Test Site (GTS) during the In-situ Stimulation and Circulation experiment (ISC) (e.g., [32,33]). Finally, we interpret our findings with respect to the hydraulic system at the GTS, and discuss the potential and limitations of the approaches for hydrological reservoir characterization.

## 2. Methods

### 2.1. Breakthrough Curves from Difference Reflection Imaging

To assess the transport of the GPR responsive tracer in an investigation volume, we compute localized GPR breakthrough curves. The local increase of conductivity, caused by a tracer presence in a thin-bed (i.e., a fracture), will increase the reflection coefficient and shift the phase of the corresponding reflection [20]. Therefore, the signal strength of such reflections should appear enhanced in GPR data acquired after tracer injection.

For quantifying changes of the signal strength, we define Regions of Interest (ROIs) and we employ a root-mean-square (RMS) measure

$$\text{RMS}(\boldsymbol{E}) = \sqrt{\frac{1}{N}\sum_{n=1}^{N}\|E_n\|^2}, \qquad (1)$$

where $\boldsymbol{E}$ is a vector of $N$ electric field data points in the ROI.

However, the signal strength increase with conductivity only holds true for a single reflector and not necessarily in a multi-reflector environment with several superpositioned reflections in the GPR data due to wave interference, which can potentially be destructive. This problem is especially pronounced for omnidirectional borehole GPR antennas, which record reflections from the entire volume surrounding the measurement borehole. Reflected waves can experience interference, even though the respective reflectors are spatially far apart. For such cases, we need to operate with difference data instead of monitoring data only. While signal strength in monitoring data is a fairly robust measure with little requirements in terms of exact reproduction of the data acquisition, successful differencing of recorded GPR data requires greater care during acquisition and processing (see e.g., [30]).

### 2.1.1. Benefits of Difference Reflection Imaging

To illustrate the benefits of difference reflection imaging, we calculated a synthetic borehole GPR response, as shown in Figure 1a. It represents hypothetical recordings with a single-hole borehole GPR setup (250 MHz transmitter and receiver antennas, 1.3 m separated from each other). We set up a simplistic case, in which the GPR images a thin-bed reflector, resembling a fracture with 0.5 mm aperture (thin-bed separation of 0.5 mm), parallel to the borehole at a radial distance of 5 m, and a second reflector, referred to as the *background reflector* (for example another borehole, fracture, or a shear zone) that is oblique to the borehole. Note that due to the azimuthal ambiguity of undirected GPR borehole antennas, the fracture and the background reflector do not necessarily need to intersect spatially. For the analytical response we assumed a Ricker wavelet as a GPR signal representation, the fracture and borehole responses were calculated through a geometrical reflection point calculation with a homogeneous GPR wave velocity in the subsurface of 0.12 m/ns.

The signal strength of the background reflection was chosen arbitrarily. From the analytical solutions of a thin-bed reflection (described for example in Deparis and Garambois [34]), we determined the reflection coefficient and the phase shift for the fracture reflection. The GPR wave frequency was chosen as 250 MHz, the fracture was water filled ($\varepsilon_r = 81$, $\sigma_f = 0.2$ mS/cm) and embedded in a host rock with a permittivity of $\varepsilon_r = 7$ and a conductivity of $\sigma_{\text{rock}} = 0.01$ mS/cm, as is reasonable for the conducted field experiments later in this work. The retrieved reflection coefficient and phase shift from this analytical calculation were then applied on the assumed Ricker-type reflection signal from the fracture. We refer to these data, acquired prior to tracer injection as the *reference data*.

After injection of a tracer, we assume the fluid in the fracture to have an electrical conductivity of $\sigma_f = 20$ mS/cm. The resulting GPR profile within the indicated ROI is shown in Figure 1b and is referred to as the *monitoring data*.

Figure 1c shows a single GPR trace within the selected ROI (as indicated Figure 1a near the (apparent) intersection of the fracture and the background reflector. The GPR trace recorded before tracer injection, is denoted as $\boldsymbol{E_r}$ (blue), and the corresponding GPR signal after injection is denoted as $\boldsymbol{E_m}$ (dashed blue). Determining the signal strength of the monitoring data RMS($\boldsymbol{E_m}$) in the selected ROI, using Equation (1), requires summation over all traces within the ROI and their window lengths. One would expect that an increase in electrical conductivity would result in an increase of RMS($\boldsymbol{E_m}$), because the impedance contrast and thus the reflection strength is increased. However, as can be inferred from Figure 1c, this is not necessarily the case, due to interference between the reflection signals of the fracture ($\boldsymbol{E_f}$) and the background reflector ($\boldsymbol{E_b}$). To showcase this,

the expected reflections from $E_b$, $E_f$ (fracture before tracer injection), and $E_f'$ (fracture after tracer injection) are presented in Figure 1c. Their respective sums make up for the reference and monitoring signals $E_r$ and $E_m$. It is apparent that the signal strength of the monitoring trace $E_m$ is reduced due to the tracer injection, which has increased the fracture reflection strength. Consequently, in Figure 1b for the monitoring calculation ($\sigma_f = 20\,\text{mS/cm}$), the signal appears to be weaker in the middle of the selected ROI, than in the reference calculation in Figure 1a ($\sigma_f = 0.2\,\text{mS/cm}$).

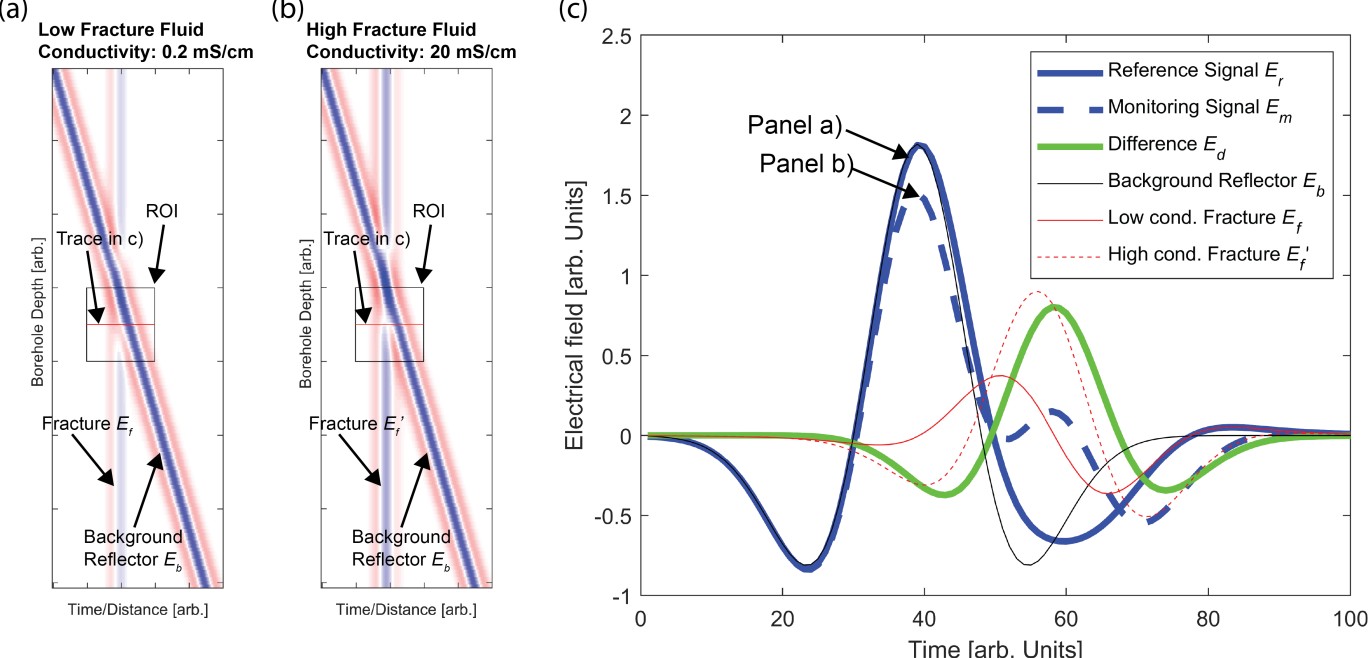

**Figure 1.** (**a**) Synthetic response for a thin-bed fracture ($E_f$, with a conductivity $0.2\,\text{mS/cm}$) and an additional background reflector ($E_b$). The box shows the region of interest (ROI), and the trace that is presented in (**c**). (**b**) Shows the ground penetrating radar (GPR) response for an increased electrical conductivity of the fracture fluid to $20\,\text{mS/cm}$ ($E_f'$). (**c**) Synthetic GPR traces within the ROI, marked in (**a**,**b**). The reference trace ($E_r$, solid blue) corresponds to panel (**a**) and the monitoring trace ($E_m$, dashed blue) to panel (**b**). The data difference $E_m - E_r$ is shown in green. Additionally, the signals expected from solely the background reflector (black) and the fracture (red low conductivity, dashed red high conductivity) are plotted. Note that the horizontal axis in (**a**,**b**) and the horizontal axis in (**c**) represent time ($E_{r,m}(t)$), but assuming a constant velocity model, this can be interpreted as the radial distance from the borehole.

### 2.1.2. Difference Radar Breakthrough Curves

The electrical conductivity of the fluid within the fracture is now varied systematically, as shown with the dashed blue line in Figure 2a. This figure shows the RMS($E_m$) of the ROI (vertically shifted to start at zero), shown in Figure 1a, as a function of the electrical fluid conductivity $\sigma_f$ within the fracture. Between 0 and $5\,\text{mS/cm}$, the values slightly decrease (minimum at $\sigma_f = 5.2\,\text{mS/cm}$) before they start to increase. The RMS of the monitoring data can therefore not be used as a suitable measure for a conductivity increase in this ROI.

This problem can be resolved by considering the data difference and calculating RMS($E_m - E_r$). As indicated with the green curve in Figure 1c, the data difference exhibits a clear signal at the location of the fracture, and it is thus not surprising that the RMS($E_m - E_r$) vs. conductivity curve in Figure 2a (orange) shows a much clearer correspondence between these two quantities that appears nearly linear. This linearity can be regarded as to some extent unexpected, because the increase in reflection strength and the change in phase due to conductivity increases is non-linear. This can be derived with the analytical calculations following Deparis and Garambois [34] and is seen in Tsoflias and Becker [20]. However, for lower conductivities the reflection coefficient is less affected while the phase is stronger

affected, and for higher conductivities the reflection coefficient is stronger affected while the phase is less affected. These two competing processes appear to result in the nearly linear behavior of the calculated difference RMS.

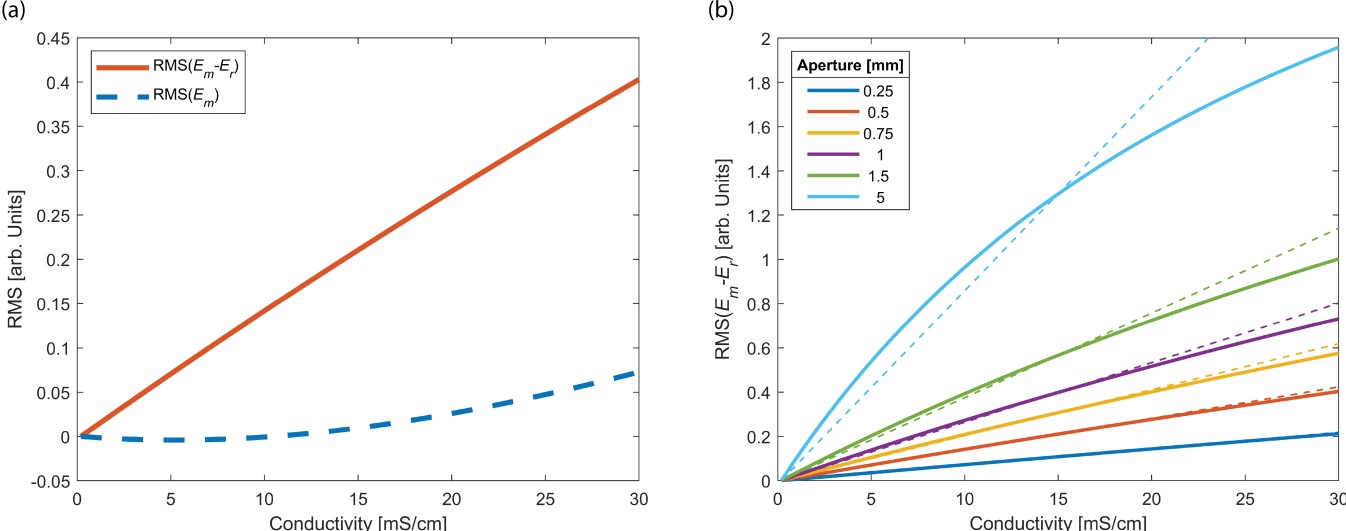

**Figure 2.** (a) Root mean square RMS($E_m$) (dashed blue, shifted to start at zero) and RMS($E_m - E_r$) (orange) values as a function of the electrical conductivity $\sigma_f$ in the 0.5 mm aperture fracture. A clear linear dependency for RMS($E_m - E_r$) can be seen. (b) RMS($E_m - E_r$) for different fracture apertures. For fractures within the sub-mm range, a quasi-linear dependency on $\sigma_f$ is visible. For comparison, a strictly linear dependency between the obtained values at 0 and at $\sigma_f = 5$ mS/cm is depicted as dashed lines. Note the different y-axis scale of both panels.

We further investigated this dependency by systematically varying the fracture aperture and calculating the corresponding RMS($E_m - E_r$) responses to conductivity increases. The results are shown in Figure 2b.

We found the RMS of the signal differences RMS($E_m - E_r$) to be quasi-linearly dependent on the electrical conductivity $\sigma$ for sub-mm fracture apertures. This inferred linearity also held true for calculations with different wave incidence angles (not shown here).

The data differencing approach, outlined in Figures 1 and 2, is also applicable to more complex scenarios including multiple reflectors. Therefore, conductivity breakthrough curves, hereafter referred to as DRBTC (difference radar breakthrough curve), can be computed for theoretically arbitrary locations by repeatedly acquiring GPR data during a tracer experiment and plotting the RMS($E_m - E_r$) values for a particular ROI as a function of time.

### 2.1.3. Limitations

We are aware that our approach to calculate synthetic GPR responses for a tracer experiment is a simplified approximation in several ways. Most importantly, we calculated the reflection coefficient and phase shift at a thin-bed reflector for a single frequency, but then applied this result on a Ricker wavelet, which is by definition composed of a broad band of different frequencies. We therefore conducted confirmation simulations, using the effective dipole approach from Shakas and Linde [35] for realistic wavelets and their reflection responses for the model geometry above. The results confirmed the quasi-linear relationship between RMS($E_m - E_r$) and the electrical conductivity in the fracture for our test cases. Further, the thin-bed equations are derived with the assumption of locally planar and parallel fracture walls, which limits the validity for strongly heterogeneous fractures.

Although the concept of DRBTC is attractive, we would like to emphasize a few important caveats. First, the repeatability of the GPR measurements has to be very high. Data differencing of field data is a non-trivial task and requires sophisticated procedures, as described, for example, in Giertzuch et al. [30,31]. Second, the absolute values of the

calculated DRBTCs are not directly convertible into electrical conductivity values. While the linear dependency allows for relative comparisons, the absolute reflection strength will be dependent for example on the fracture geometry and aperture. Hence, a local calibration measure would be needed to obtain absolute values. Finally, an inherent problem is caused by the azimuthal ambiguity of the GPR antennas. In the presence of several distinct flow paths in the investigation volume, it is possible that the electrical conductivity changes simultaneously at different locations but similar radial distances, and the RMS($E_m - E_r$) values of a particular ROI will represent only the summed effects of the different changes. This will complicate the interpretation of the results.

### 2.2. Average Flow Path Velocity Distribution

A simple option for obtaining tracer flow velocity information, is to divide the Euclidean distance between an injection and an observation location by the time difference between the injection and the appearance at the observation location. However, this underestimates the traveled distance and therefore velocity, for all non-straight flow connections. If only sparse information (e.g., sensors in borehole intervals) is available, it will be challenging to delineate tortuous flow paths that deviate from the Euclidean distance, and hence to calculate more representative tracer flow velocities.

From time-lapse GPR measurements it is possible to delineate deviated flow paths. This allows for a tracer experiment analysis that is not limited to localized observations points. In a two-step procedure the average tracer velocity along a flow path can then be estimated. First, the tracer flow paths and their lengths need to be reconstructed, and then the propagation velocities along those paths can be determined from the temporal information in the GPR data.

### 2.2.1. Flow Path Reconstruction

A comprehensive description on flow path reconstruction using GPR data from saline tracer experiments is provided in Giertzuch et al. [31]. Here, we briefly summarize the key steps involved. From time-lapse reflection GPR surveys, the tracer reflection signal can be identified by applying a data differencing approach (e.g., [26,30,36]). From the different acquisition times during the time-lapse survey, the tracer positions in the reflection profiles can be associated with an arrival time $t$. However, due to the undirected borehole antennas, these reflection signals cannot unambiguously be associated with a spatial location. Instead, after processing, identified tracer reflections at a time step $t$ can have occurred within a toroid around the measurement borehole. This is depicted in Figure 3a, where the diameter of the toroid indicates the radial distance of the tracer reflection from the borehole, and its "thickness" is determined by the size of the reflection. If data from at least two boreholes are available, then the spatial location of possible tracer reflections can be better constrained from the intersection of the two resulting toroids. This intersection in 3D can be understood as a volume, in which tracer reflections are most likely to have occurred. In Figure 3a only a single intersection is depicted that results in a relatively large volume of possible tracer location, but also two intersections could occur, hence defining two smaller but disjoint volumes (see bottom of Figure 3a. To delineate the entire tracer propagation over time, the intersections for every time step $t$ have to be calculated, and their combination finally defines the regions of possible tracer locations. This is shown in Figure 3b, where a cross section along the borehole plane is depicted. It is a similar scenario as in the field example shown later in this contribution. The identified possible tracer locations lie between the two boreholes and the possible reflection volumes are associated with the time steps $t$ at which the tracer signals were identified in the time-lapse reflection survey. Clearly, even with two sets of GPR borehole measurements from different boreholes, there still exists a considerable uncertainty concerning the spatial location of the flow paths. Nevertheless, the procedure offers spatial and temporal information that allows the dynamic tracer propagation to be further evaluated.

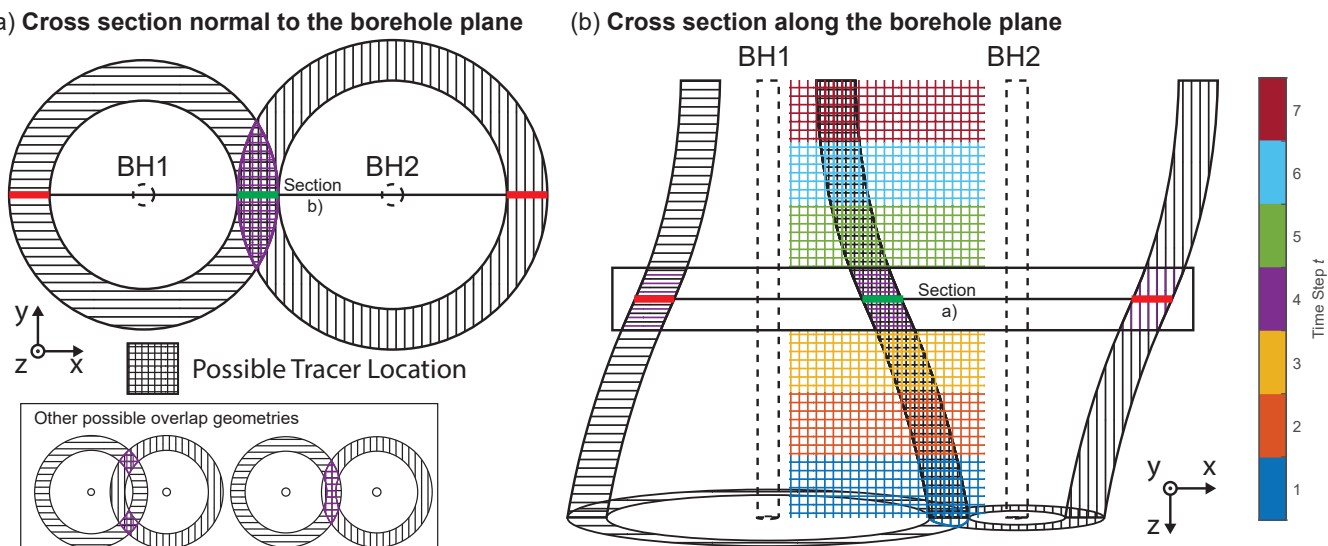

**Figure 3.** Method to reconstruct tracer flow path. Reflections can arise from positions radially symmetric to the borehole axis. Two boreholes allow to find intersections and constrain the location of the tracer, indicated by the squared pattern. (**a**) Cross section normal to the borehole plane. (**b**) Cross section within the borehole plane at time step $t = 4$. The colors indicate reflections identified during the associated time steps $t$.

### 2.2.2. Average Flow Path Velocity Calculation

The flow path reconstruction approach does not only yield the flow path geometry, but also associates locations within the reconstructed volume of possible tracer reflections with the time at which the tracer appeared. This can be exploited to compute average tracer flow velocities.

In a case of very high spatial and temporal resolution in the time-lapse GPR measurements, one could compute almost continuous spatio/temporal gradients, using a procedure similar to those employed for backtracing rays through a travel time field obtained from a finite-difference Eikonal solver (e.g., [37]). However, a GPR tracer experiment typically includes only a limited number of repeated GPR profiles (i.e., a limited number of time steps) due to the necessary acquisition time, and reconstructed possible reflection points can show tracer large propagations during each time step and are subject to significant location uncertainty. This is illustrated in Figure 4. For the sake of simplicity, it shows a 2D scenario, but an extension to 3D is straightforward. The tracer spreads from the initially tracer filled source (blue) to the target area (red). The associated time steps $t$ during this propagation are shown with different colors.

To retrieve flow path lengths and tracer velocities, we applied a routing algorithm that operates on discretized points of possible tracer reflections and is based on nearest neighbor calculations between time steps. Figure 4 shows an example for a possible reflection point, where the tracer was visible at time step $t = 7$. The algorithm calculates the distances to all points that were associated with time steps of $t < 7$ and the point with the closest distance is chosen. This procedure is repeated from this point until a point within the first time step is reached. This allows to compute reasonable propagation distances along the tracer flow paths. In Figure 4, the hydraulic connection that was determined this way is depicted with a solid arrow. The covered distance (cumulative distance for the identified connections) for this case differs significantly from the direct connection (solid and dashed arrow), due to the curved nature of the shown tracer flow. The procedure is robust also for multi-directional flow with different velocities as highlighted with the smaller dotted arrow in Figure 4: for the point that was added during $t = 3$ (yellow), the connection does not necessarily include a point from $t = 2$ (orange), as the connection is always determined with respect to all points $p_{<t}$ and not $p_{t-1}$.

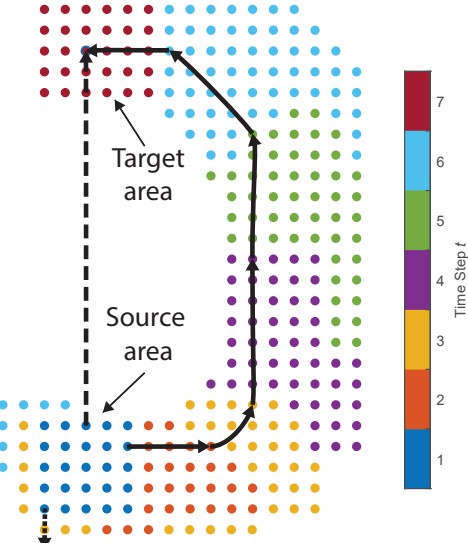

**Figure 4.** 2D schematic of the routing algorithm. The solid black arrow is the reconstructed flow connection from the blue source area ($t = 1$) to the red/blue point in the red target area ($t = 7$). The dashed arrow depicts the direct connection. Smaller dotted arrow: flow from source area to a different target point. Different colors indicate different time steps $t$.

### 2.2.3. Limitations

It must be noted that the routing algorithm will always result in the shortest possible connection through the temporal evolution of the tracer point cloud and is therefore prone to "cutting corners". This problem is particularly pronounced for low temporal and spatial resolutions, and it will therefore underestimate the actual distances and velocities. However, it is still a clear improvement compared with the straight connection assumption.

In essence, the described routing algorithm relies on the closest distance towards the tracer propagation front of the previous time steps. Points $p_t$ which do not belong to a propagation front can never be connected to a point in the next time steps $p_{>t}$, because the distance to a point on the front will always be smaller. For this reason, the locally covered distances within a certain time step will be likely underestimated for any point that is not part of the propagation front. This would affect local (intra-time-step) velocities, which are therefore not reasonable to calculate. The relation between the cumulative distances and the total time since time step $t = 1$ will, however, remain intact. Thus, the procedure can only be used to calculate the average velocity along sufficiently long flow paths, which is not straightforwardly translatable into the local tracer flow velocity.

We also want to emphasize that the resulting tracer reflection point connections that are determined in this procedure do not represent the actual physical flow paths along the true fracture network, as they are below our resolution limit. Instead, the procedure only offers a possibility to calculate reasonable flow path distances along the reconstructed 3D tracer propagation.

## 3. Application to Field Data Set

### 3.1. ISC Experiment at the Grimsel Test Site

The Grimsel Test Site (GTS) is operated by NAGRA, the Swiss Cooperative for the disposal of radioactive waste, and has been used for a range of research projects (www. grimsel.com (accessed on 11 May 2021)). This underground rock laboratory is located in a weakly fractured rock mass in the central Swiss Alps.

The experiments documented in this contribution were part of the In-situ Stimulation and Circulation (ISC) experiment [32,33,38]. The ISC experiment was carried out to investigate the seismo-hydromechanical processes during the creation of a geothermal reservoir. It included two stimulation phases, featuring hydraulic shearing and hydraulic fracturing that were conducted in February and May 2017, respectively [33]. A comprehensive characterization for the relevant part of the GTS was crucial for a detailed understanding of the stimulation effects in the subsurface and a subsequent transfer of knowledge into reservoir creation. Therefore, additionally to the monitoring performed during the stimulations, pre- and post-stimulation characterizations were carried out with various geological, hydrogeological, and geophysical methods. Aside from GPR, as described here, by Giertzuch et al. [30] and by Doetsch et al. [39], extensive hydrologic testing was performed, tracer tests were run, and borehole logs were acquired, e.g., [7,12,13,38,40–43].

The ISC experiment was conducted in the southern part of the GTS, where the overburden is approximately 480 m. Figure 5 shows a geological model of the relevant part of the GTS, based on the work of Krietsch et al. [38]. It includes two main shear zones S1 and S3, whereby S3 can be subdivided into sub-shear zones S3.1 and S3.2 (shown green in Figure 5). The fracture density in the host rock varies between 0 and 3 fractures per meter and increases to >20 fractures per meter between the S3 shear zones [38]. Doetsch et al. [39] found a decrease in seismic velocity between the S3 shear zones and were able to link this to a known increase in fracture density in this region [12,38].

The S3 shear zones intersect the AU tunnel (shown blue in Figure 5) between the geophysical monitoring boreholes (GEO1 and GEO3), which were used for the GPR surveys presented here. In total, 15 boreholes were drilled in the project volume to characterize the subsurface conditions, but Figure 5 shows only those relevant for the GPR tracer experiments. Besides the GEO1 and GEO3 boreholes, this includes the injection borehole (INJ2, green), and three pressure monitoring boreholes (PRP1-3, magenta). Straddle packers were installed around the borehole intervals indicated in red to isolate permeable fractures. The tracer injection interval (INJ2.4, borehole depth 22.89–23.89 m) is indicated as a red sphere.

The ISC experiment volume and its fracture network are fluid-saturated. Fluid flow was shown to be primarily fault-controlled in well testing experiments by Brixel et al. [13], who found permeability to be strongly decreasing (from $10^{-13}$ m$^2$ to $10^{-21}$ m$^2$) within 1 m to 5 m from the fault cores. Brixel et al. [12] determined the equivalent hydraulic apertures of single fractures in the rock volume to range from 2 to 130 μm, with a mean of 30 μm. However, this calculation relies on a parallel plate assumption with smooth fracture walls, hence locally the true fracture apertures can diverge significantly from this. The S3 shear zones appear to have feature extension fractures that are linking the two S3 shear zones and create highly permeable cross-fault connections as shown by Brixel et al. [13]. Additionally to the fault-controlled nature of the flow system, a drainage effect of the AU tunnel plays a key role for fluid flow at the ISC test site [7,44]. From dye and saline tracer tests in the INJ2.4 interval, connections towards PRP1.3, PRP2.2 and the AU tunnel with relatively fast breakthroughs in that order (within several minutes to a few hours) are known [7,30,43,45].

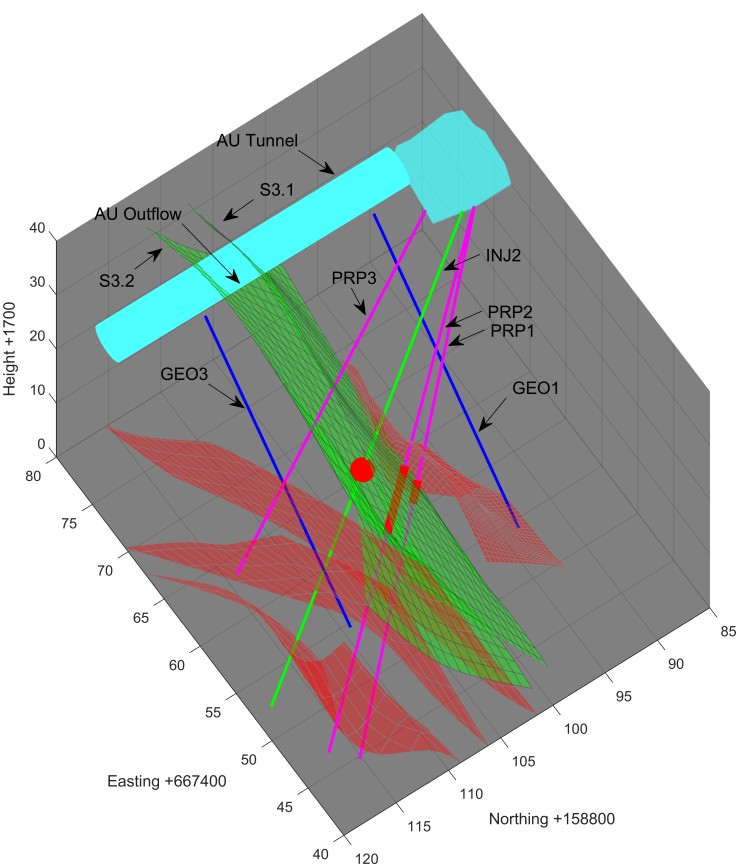

**Figure 5.** Geological model of the experiment volume with relevant structures and boreholes. The S1 and S3 shear zones are shown in red and green, respectively. The GPR survey was performed in the blue GEO boreholes. The salt tracer injection point is indicated as a red sphere. Monitoring intervals in the PRP boreholes are indicated in red.

### 3.2. Tracer Experiments

Two GPR tracer experiments were conducted in November and December 2017. Both experiments used a tracer of the same salinity that was injected at the same injection location, and they had a similar GPR monitoring setup. The formation water showed an electrical conductivity of approximately 80 µS/cm, and the saline tracer had an electrical conductivity of approximately 60 mS/cm. Before and after tracer injections, formation water was injected continuously with constant pressure for several days at the injection interval to ensure a steady flow state, resulting in a constant injection flow rate of approximately 2 L/min. Conductivity dataloggers were connected to the AU outflow, and to the intervals PRP1.3 and PRP2.2 (see Figure 5). The maximum electrical conductivity measured with the dataloggers during the experiments was found in PRP1.3 at below 30 mS/cm and the AU outflow showed maximum conductivities at below 5 mS/cm. For the first experiment (*Experiment 1*), 100 L of saline tracer were injected over 50 min, while for the second experiment (*Experiment 2*) the volume was doubled to 200 L, and the injection lasted 100 min. Experiment 2 was performed during an ongoing heat tracer survey. Although the hydraulic properties of the individual flow paths appeared to be affected by this survey [43], the geometry of the flow paths can be assumed to have remained unchanged.

### 3.3. GPR Data Acquisition and Processing

The GPR data were acquired as single-hole reflection GPR data sets, recorded from separate (but parallel) boreholes. The survey during Experiment 1, featuring 100 L of saline tracer, was conducted in the GEO3 borehole, while the survey during Experiment 2, featuring 200 L of saline tracer, was conducted in the GEO1 borehole (see Figure 5). In all

of the described GPR experiments, Malå (Malå Geosciences, Malå, Sweden) 250 MHz GPR borehole antennas were used with the Malå CU2 control unit.

The antenna separation was held constant for both surveys at 1.76 m, and a trace was recorded every 10 cm along the borehole. This distance was measured with a trigger wheel, and the measurements were recorded while pulling the antenna array upwards, to ensure constant cable tension. The data acquisition during Experiment 1 lasted for 5 h, but it was partly interrupted due to empty antenna batteries. In total, 34 usable reflection profiles were recorded. The data acquisition during Experiment 2 was carried out as a combined reflection and transmission survey. However, only data from the GEO1 single-hole reflection survey are considered here (see Giertzuch et al. [31] for a combined analysis of reflection and transmission surveys). In total, 74 usable reflection profiles over the course of approximately 7 h were recorded, again partly interrupted for battery recharging.

GPR data processing was performed with the workflow developed by Giertzuch et al. [30], and is described for the two experiments presented here in Giertzuch et al. [31]. The key processing steps included: Frequency filtering, temporal and spatial trace alignment, first arrival and noise suppression, time gain, data differencing, migration, and temporal smoothing.

### 3.4. Breakthrough Curves from Difference Reflection Imaging

### 3.4.1. ROI Selection

We selected ROIs in the difference profiles, as seen in Figure 6 for the Experiments 1 and 2, which were monitored from GEO3 and GEO1, respectively. To compare with the sensor data, we chose a ROI around the estimated reflection position of the PRP1.3 interval, and additionally selected a region between the injection point and the AU tunnel. This area was described by Giertzuch et al. [31] with relatively good tracer contrast around a borehole depth of approximately 13 m. It was therefore chosen as a ROI for the BTC calculations, and is henceforth referred to as *ROI 2*. For Experiment 2, we could further select ROIs for two separate flow paths from the injection point towards PRP1.3 (labeled PRP1.3 A and PRP1.3 B). These ROIs were defined based on the observed separated flow paths in the difference profiles towards the PRP1.3 interval [31].

Selection of a ROI closer to the AU tunnel turned out to be unfeasible, because the difference profiles tended to appear very noisy at the vicinity of the tunnel. Likewise, it was also unfeasible to select ROIs over the injection interval and/or the PRP2.2 borehole interval. Both of these overlap in the reflection profile (as presented in Giertzuch et al. [31], because they share a similar radial distance to the borehole. From the conductivity sensors, we know that the PRP2.2 interval shows a strong breakthrough shortly after the PRP1.3 interval. However, in the GPR difference images, increases in PRP2.2 and the injection point could not be distinguished due to azimuthal ambiguity in the antenna radiation.

### 3.4.2. DRBTC Results

GPR reflection amplitudes are strongly affected by variations in fracture geometry and aperture. Therefore, a comparison of absolute values in separate regions of the profile is not useful. This can also be seen in the analytical calculations for different apertures in Figure 2b. Varying noise levels for DRBTC calculations in the chosen ROIs are observed. Instead of comparing absolute values, we therefore normalized the DRBTCs towards their maxima, subtracted the pre-injection baseline noise, and can thus compare relative differences. To make a comparison with the BTC measurements from the conductivity sensors, we applied the same normalization here, instead of the commonly applied normalization towards the original tracer conductivity.

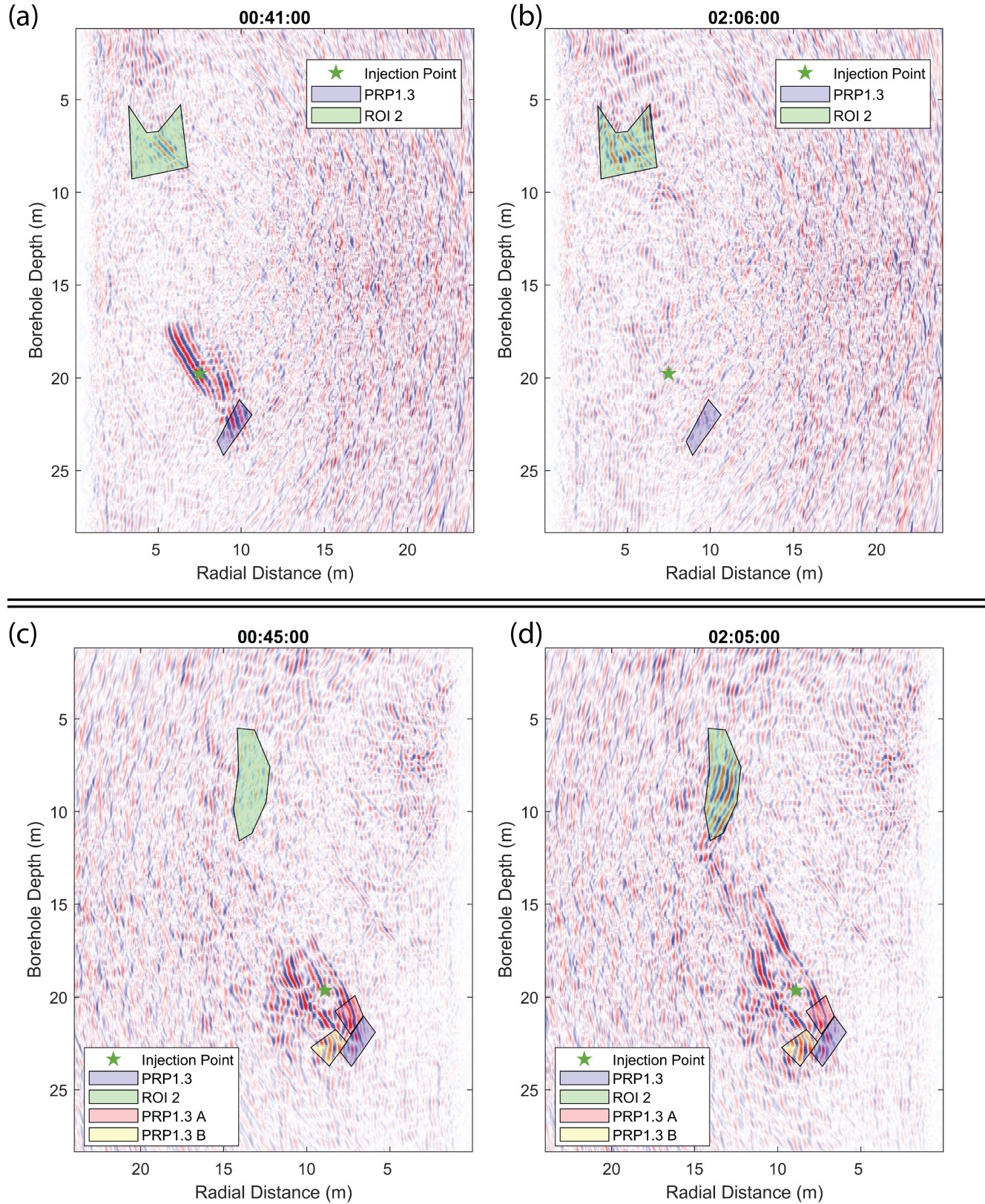

**Figure 6.** Definition of regions of interest (ROIs) in the GPR reflection surveys. (**a**,**b**) show different time steps of Experiment 1 monitored from GEO3 and the selected ROIs. (**c**,**d**) show different time steps of Experiment 2 monitored from GEO1 and the selected ROIs. Figure adapted from Giertzuch et al. [31]. Note that the *x*-axis with the distance from the borehole has an opposite direction for the two experiments, such that the geometry is better comparable.

**Experiment 1**

Figure 7a shows the calculated DRBTC in comparison with the measured conductivity sensor BTC for the PRP1.3 interval. The DRBTC shows an earlier breakthrough, an earlier peak ($t = 35$ min), and decreases much faster down to noise level than the sensor data, which exhibit a long tail. The sensor BTC decreases relatively quickly after reaching the peak at $t = 64$ min, but around $t = 90$ min the decrease slows down and forms a long tail. For steady state flow, such a BTC shape can be indicative that multiple flow paths connect the injection interval with PRP1.3. The BTC shape seems to indicate at least two arrivals and therefore two flow paths, an early one forming the peak and quick decrease afterwards (due to the transported formation water that was injected after the tracer), and a second one which seems to arrive later (approximately $t = 90$ min) and thus slowing down the decrease (by still transporting highly conductive tracer). Additional flow paths to explain the long tailing in the curve are possible, as can be expected for fractured flow systems (e.g., [46,47]). The form of the sensor BTC is comparable with results from Kittilä et al. [43] and the multi-flow-path assumption is also in agreement with this publication, where data for a dye tracer that was injected during this experiment were evaluated. The DRBTC appears to match the shape of the beginning of the sensor BTC and might therefore sense the early flow path towards PRP1.3 only, instead of the actual PRP1.3 sampling interval. This would also explain the earlier breakthrough in the DRBTC data, because the DRBTC would effectively measure over a location between the injection point and the PRP1.3 interval.

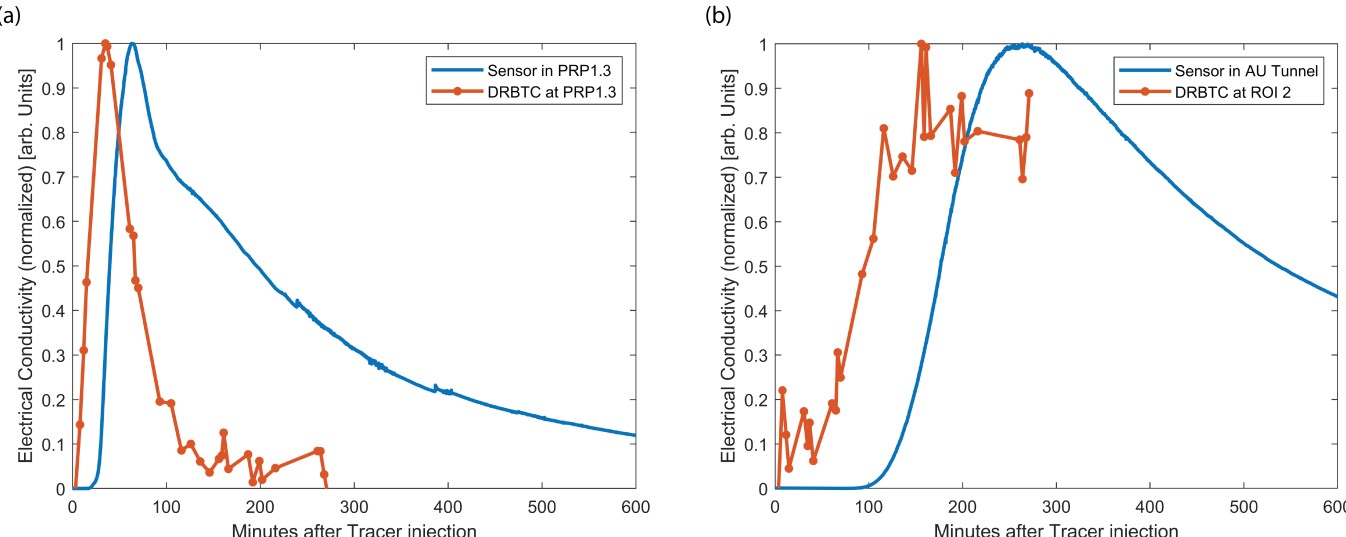

**Figure 7.** Comparison of the difference radar breakthrough curves (DRBTC) and sensor breakthrough curve (BTC) during Experiment 1. (**a**) Shows the data for the PRP1.3 interval and the accordingly defined ROI. (**b**) Shows the DRBTC for ROI 2 and the conductivity sensor BTC for the AU tunnel outflow (which is not at the same location as ROI 2).

Figure 7b shows the corresponding results for ROI 2 and compares it with the sensor data that were acquired in the AU tunnel. The AU tunnel sensor was the closest observation location to ROI 2 for conductivity measurements, but both locations are separated by approximately 9 m. Both curves (sensor BTC and DRBTC) appear to have a similar shape, but they are shifted in time. Because the tracer propagated from ROI 2 further towards the AU tunnel, this time difference is therefore expected. The signal-to-noise ratio (SNR) for the DRBTC in ROI 2 is very low, but the breakthrough can still be recognized. The ROI 2 curves are generally broader with a less pronounced peak than in the PRP1.3 data (Figure 7a).

**Experiment 2**

Figure 8a shows the calculated DRBTC in comparison with the measured sensor BTC for the PRP1.3 interval. Both curves are in good agreement until about 160 min, when the sensor BTC further decreases while the DRBTC remains elevated and/or reaches an in-

creased noise level. Their agreement indicates that both, the sensor and the DRBTC, sensed a similar region during Experiment 2. Both curves rise strongly in the beginning, then the increase slows down around 50 min, before reaching the peak at around $t = 120$ min. The sensor BTC shows a steep decline to lower levels compared to the DRBTC. This supports the assumption that especially the later part of the DRBTC can be affected from the azimuthal ambiguity and noise. Similar to Experiment 1, the DRBTC appears slightly offset to earlier times in comparison to the sensor BTC. Due to the increased tracer volume and injection duration (200 L and 100 min vs. 100 L and 50 min in Experiment 1), the curves are generally broader. The dual-flow-path shape that was observed in the sensor data of Experiment 1 does not appear to be pronounced in this case. However, this can likely be attributed to the longer injection time. For this reason, it is likely that the first flow path did not yet transport formation water during the tracer arrival along the second flow path.

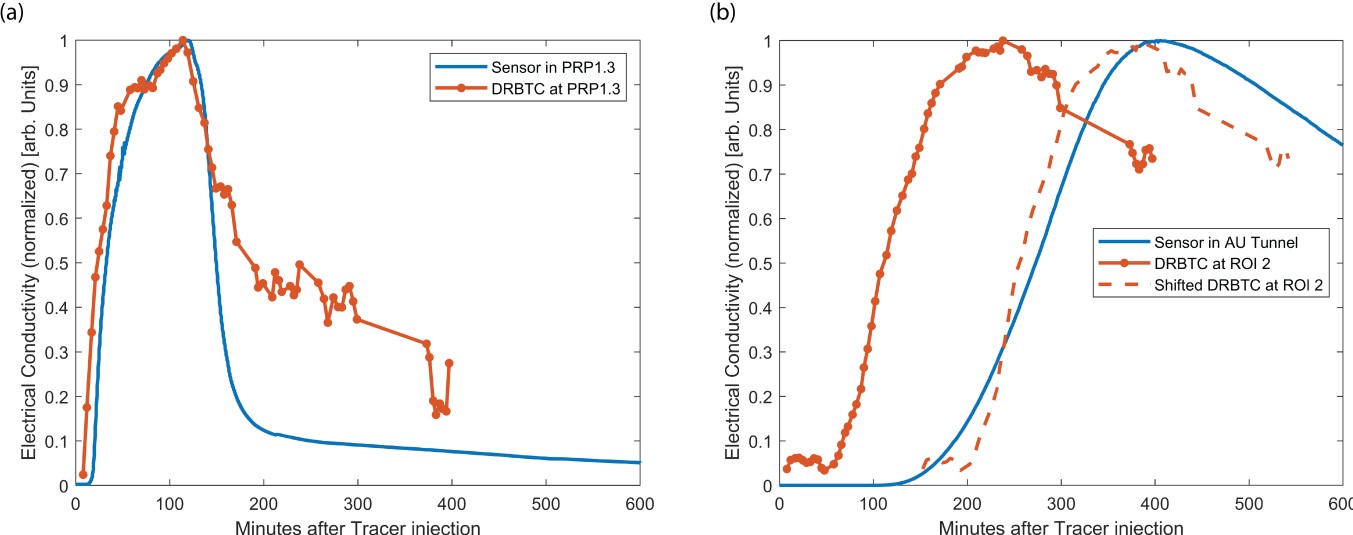

**Figure 8.** Comparison of the DRBTC and sensor BTC during Experiment 2. (**a**) Shows the data for the PRP1.3 interval and the accordingly defined ROI. (**b**) Shows the DRBTC for ROI 2 and the sensor BTC for the AU tunnel outflow. Additionally, the DRBTC for ROI 2 has been shifted to the extrapolated arrival time at the AU tunnel from the velocity field of Experiment 2 in Section 3.5.

The elevated noise level for later times in the DRBTC can likely be attributed to two factors. First, the GPR data acquisition system stability tended to decrease over time, which generally increased the noise in the differenced data, which can be critical for ROIs with a low SNR. Second, the problem of radial ambiguity of the reflections around the borehole makes it possible that parts of the tracer propagated in regions with similar radial distances as PRP1.3, but different spatial locations.

Figure 8b shows the results for ROI 2 (see Figure 6). Again, the DRBTC is compared to the sensor data from the AU tunnel. The DRBTC breakthrough time and maximum are reached earlier than those in the sensor BTC, due to the different positions. The SNR of the DRBTC in ROI 2 is high, hence it was possible to acquire DRBTC data of good quality for a region that was formerly unavailable for hydraulic data acquisition. For Experiment 2 it was possible to calculate the average flow path velocity distribution in Section 3.5. This allowed shifting the DRBTC for ROI 2 by the extrapolated time difference (145 min) between ROI 2 and the AU tunnel, which is shown as a dashed curve in Figure 8b. The time-shifted DRBTC and the AU tunnel sensor BTC show a good temporal agreement. The difference in shape is expected due to dispersion and flow heterogeneity. Note that this procedure was not applied to the DRBTC results from Experiment 1 (Figure 7b), because the temporal resolution of the Experiment 1 data did not allow the calculation of the average flow path velocity distribution.

To further investigate the dual-flow-path assumption, the two separate flow paths towards the PRP1.3 interval that were identified in the difference profiles from Experiment 2 were used as ROIs. We calculated the distinct DRBTCs for both paths with the ROIs shown in Figure 6 and the results are presented in Figure 9. The overall shape of the curves appears similar, however, the timing is different. We can infer that the upper flow path in the GPR difference image (PRP1.3 A) is reached earlier than the PRP1.3 ROI, while the lower flow path (PRP1.3 B) shows a slightly later breakthrough than the PRP1.3 ROI. Therefore, the PRP1.3 interval appears to be connected by at least two flow paths, of which one arrives quickly, while the other takes more time.

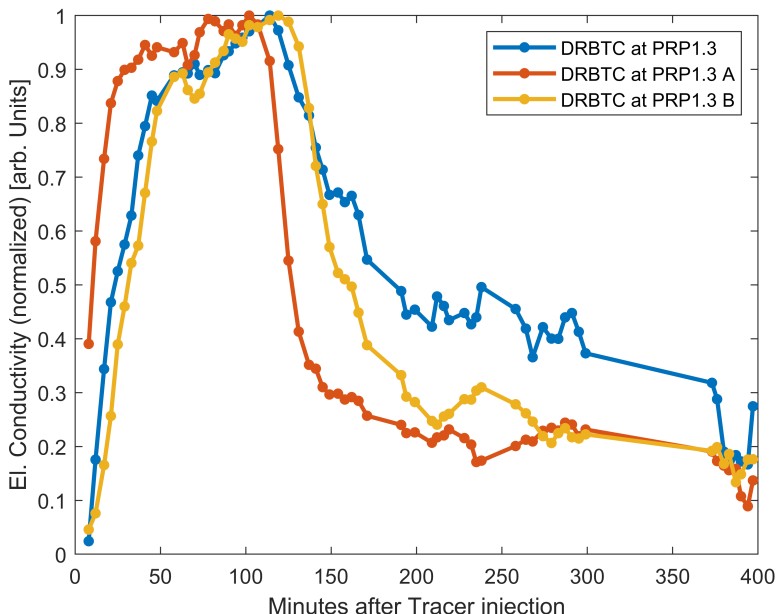

**Figure 9.** DRBTCs for the two inferred flow paths A and B towards PRP1.3, along with the PRP1.3 DRBTC.

### 3.5. Average Flow Path Velocity Distribution

Our method to obtain a flow path averaged tracer velocity distribution in 3D was applied to the tracer flow path reconstruction from Giertzuch et al. [31], which allowed for the delineation of the 4D tracer propagation in the Experiments 1 and 2. There, the tracer propagation towards the AU tunnel was shown to have mostly occurred along the S3.2 shear zone.

For geometric reasons, the 3D reconstruction from Giertzuch et al. [31] suffers from a strong positioning uncertainty normal to the borehole plane, as illustrated in Figure 3a. Hence, the propagation from the injection point towards the point cloud positions in the first time step after tracer injection is blurred strongly in this direction. We could therefore not calculate the velocities from the injection point towards the points within the first time step after tracer injection. Instead, only the velocities from the point cloud of the first time step ($t = 8\,\mathrm{min}$ after tracer injection) to the later ones were calculated. This way, the spatial ambiguity remains similarly expressed and has only a little effect on the velocity field. For this reason, the point cloud of the first time step is also not shown in the results.

We applied our routing algorithm to the Experiment 2 data only, as the temporal resolution of Experiment 1 was inferior and did not allow for a successful application. Figure 10a shows the calculated cumulative distance from the point cloud in the first time step $p_{t=8\,\mathrm{min}}$ to each tracer position $p_{t>8\,\mathrm{min}}$. Furthermore, an example of a reconstructed path for flow from the point cloud in $t = 8\,\mathrm{min}$ towards a point in the selected ROI 2 from Section 3.4.1 (green sphere) is shown. The retrieved cumulative distances and the example flow path appear reasonable, when compared to the 3D geometry.

Figure 10b shows the calculated average tracer velocities over the reconstructed flow paths from the point cloud at $t = 8$ min to tracer positions $t > 8$ min. Not all points are shown for the sake of image clarity. This average velocity appears to be low around the initially filled volume of time step 1, and increases towards and within the S3.2 shear zone. Then, the velocity decreases again towards the AU tunnel. The maximum flow path averaged tracer velocity was found to be 0.0036 m/s. The tracer velocities from the injection point towards PRP1.3 and PRP2.2 could not be resolved, because these locations lie within the point cloud of time step 1 (due to the location uncertainty, which is large also because the direction from the injection point towards PRP2.2 is approximately normal to the borehole plane). However, the velocities over these connections can be assumed to have exceeded those of the later time steps around this area. For comparison: the Euclidean distance between the injection interval and PRP1.3 is ~4.5 m and the breakthrough occurred after only ~20 min according to the conductivity sensor in Experiment 2 (see Figure 8), hence the velocity there should be $\geq$0.00375 m/s.

The Euclidean distance between the injection interval and the AU tunnel outflow is $\approx$21 m and the breakthrough occurred after approximately 150 min in Experiment 2 (according to the conductivity sensor, see Figure 8b), hence the average velocity over this connection would be $\geq$0.0023 m/s. The cumulative distance along the flow paths towards the AU tunnel could not be calculated by the algorithm, as the point cloud did not extend this far. However, an estimate can be made by assuming a direct connection from a point closest to the AU tunnel (end point) in the reconstructed tracer point cloud (=2.6 m), and adding this to the retrieved cumulative distance for the end point (=20.8 m). Because the distances were calculated from the point cloud in time step $t = 1$ and not from the injection point, also this distance needs to be accounted for. The distance from the injection point to the reconstructed starting point on the flow path towards the end point is 1.8 m, thus the total distance from the injection location along the reconstructed flow path to the AU outflow is 25.2 m. Divided by the breakthrough time $\approx$150 min in the sensor BTC in the AU tunnel, we can calculate an average velocity over this flow path of $\approx$0.0028 m/s.

Because the tracer propagation from the injection point towards the AU tunnel was found to be relatively straight (see Figure 10a and Giertzuch et al. [31]), the difference between the Euclidean distance ($\approx$21 m) and the reconstructed distance (25.2 m) is small. In general, the calculated average flow path velocities and distances in Figure 10 can be assumed to be a better estimate than a direct connection assumption, while still underestimating the true values.

Additionally, we were able to calculate the expected difference in tracer arrival time between ROI 2 (see Section 3.4.1) and the AU tunnel. To this end, we started the algorithm at the time step at which ROI 2 was tracer filled (87 min after tracer injection), and calculated the cumulative distance (=7.5 m) and according average velocity (=0.0012 m/s) from this region to the end point. We could thus extrapolate the time from the end point towards the AU tunnel (by the same direct connection assumption as above), and determine the expected time difference between ROI 2 and the AU tunnel as 145 min. Figure 8b displays the DRBTC for ROI 2 shifted about 145 min and shows good accordance with the sensor BTC, measured in the AU tunnel. It should be noted that the change in the shape is expected due to diffusion and flow path heterogeneity.

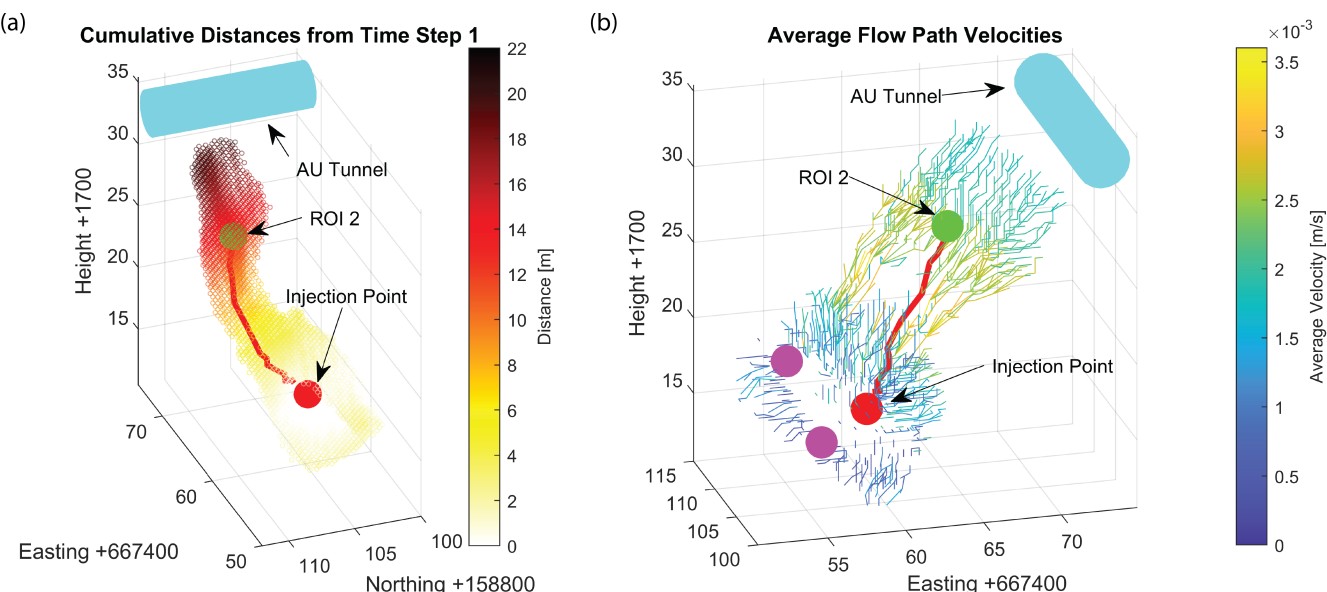

**Figure 10.** (**a**) Cumulative distances from the first time step and exemplary flow connection calculated with the routing algorithm towards a point in ROI 2 (green). (**b**) Various resulting flow path trajectories determined with the routing algorithm that are color-coded with their respective average velocity. Note that these flow paths do not represent the physical flow paths through the fracture network, but are only the reconstructed connections in order to determine apparent flow path distances and the average velocity along these. The spheres indicate the injection location (red), the PRP1.3 and PRP2.2 intervals (magenta), and a point in ROI 2 (green). As a reference, the AU tunnel is shown in cyan.

## 4. Discussion

### 4.1. Breakthrough Curves from Difference Reflection Imaging

#### 4.1.1. Difference Radar Breakthrough Curves

The comparison of the DRBTCs with the concentrations measured with the sensors in the PRP1.3 interval and at the AU tunnel indicated a relatively close agreement in shape and timing between the two methods for both experiments. In the Experiment 2 data, it was possible to differentiate two flow paths (A+B) towards PRP1.3. This provided hints to features that were previously undetected by the borehole sensors. The DRBTC method, therefore, appears to offer reasonable potential for flow and transport characterization in fractured rock.

The acoustic and optical televiewer borehole logs in the injection borehole only showed a single fracture at an angle consistent with a connection of the injection interval to the PRP1.3 interval [38]. Because the GPR difference images and the DRBTCs presented here strongly suggest at least two connections, the question arises whether these connections are flow channels within a single fracture or separate fractures, out of which one does not directly cross through the injection borehole. From the distance between the visible flow paths (A + B) in Figure 6c,d, the fracture angle in the borehole logs, and the radial symmetry assumption for the GPR antenna, we found it geometrically unreasonable that flow channeling within a single fracture can be the cause. Therefore, we conclude that the presence of separate fractures is likely. As no apparent candidate for the second fracture in the injection interval could be identified in the borehole logs, the flow path separation into two fractures probably occurred outside the injection borehole. This highlights the inherent limitation of acoustic and optical borehole logs that can only characterize a certain fraction of the overall existing fractures in the subsurface, which will, for instance, influence fracture density estimations. GPR can reduce this uncertainty (also without a tracer fluid) if the fractures are favorably oriented [48,49]. While borehole logs tend to miss fractures that are (quasi-) parallel to the borehole, GPR measurements fail to resolve fractures normal to the survey borehole. This highlights the complementary nature of these two methods.

The Experiment 1 GPR profiles do not seem to display separate flow paths between the injection point and PRP1.3, as seen in Figure 6 and further discussed in Giertzuch et al. [31]. This can most likely be explained by the different GPR acquisition boreholes and the resulting reflection angles of the fractures for both experiments. Geometrically, the strongest reflection for the fractures will be detected at lower borehole positions in Experiment 1, but at higher borehole positions in Experiment 2, due to the reflector orientations. The injection interval and PRP1.3 are closer to the bottom of the borehole. Consequently, the illumination from favorable reflection angles is better in Experiment 2 (from the GEO1 borehole) than in Experiment 1 (from the GEO3 borehole). This observation was verified in unmigrated difference GPR images. According to the comparison with the PRP1.3 sensor BTC, we assume that the DRBTC in Experiment 1 only senses flow path A, but not flow path B.

There are two main differences between the two Experiments 1 and 2: First, the amount of tracer and hence the injection duration was doubled (200 L and 100 min for Experiment 2). Second, Kittilä et al. [43] have described that the expression of flow path preferences and the arrival times were influenced by an ongoing heat tracer survey. The differences between the experiments are seen in both, the sensor BTCs and the DRBTCs, especially in the PRP1.3 interval. Due to the increased injection time in Experiment 2, the tracer along flow path B is likely to have arrived while still high conductive tracer fluid was transported along flow path A. By contrast, in Experiment 1, post-tracer formation water was probably transported earlier into the interval along flow path A, which lead to the decrease in conductivity before more conductive tracer arrived along flow path B, which could explain the observed shape of the BTC. Due to possible changes from the heat tracer survey in Experiment 2, also the early flow path A could be more pronounced and the later flow path B less pronounced compared to Experiment 1.

### 4.1.2. DRBTC Limitations

As already indicated before, multiple limitations of the DRBTC method have to be addressed. Foremost, every limitation that is already inherent in the GPR difference reflection imaging method is inherited for the DRBTC calculations. Therefore, different profile areas will resolve the data with different SNR, dependent for example on fracture geometry, but also related to the ambiguity in localization due to the antennas' radial symmetry. Depending on the flow path geometry, this can either have no impact on the data at all, or can completely rule the data useless (in case of radial flow around the borehole). Primarily for this reason, DRBTCs could not be calculated adequately for the injection and PRP2.2 interval. However, it should be noted that also traditional borehole sensors can suffer from ambiguity: If larger borehole intervals are isolated for sensor measurements, the results can be a combination of multiple fractures and flow paths.

Additionally, as the DRBTC only provides relative information, there is currently no straightforward way to draw conclusions on the actual tracer concentrations and compare the absolute data of different ROIs. This problem could potentially be overcome by performing site specific calibration measurements, as described in Becker and Tsoflias [21]. However, this could not be applied in our case, because saturating the research volume with fluids of different electric conductivities was found unfeasible for the large connected natural fracture network at GTS.

Furthermore, the SNR of DRBTCs is strongly inferior to that of classical conductivity sensors and it even decreases over time due to instabilities in the GPR data acquisition. Therefore, for the experiments presented here, late time information, including the tailing of the DRBTCs, could not be retreived. Thus, even though a common way of displaying and analyzing BTC data, it was not reasonable to investigate the DRBTCs on a logarithmic scale. However, this difficulty will be differently expressed at different experiment sites, ROIs and GPR setups, and therefore multiple approaches to overcome this problem are possible. A set of directional borehole GPR antennas with high system stability could strongly enhance the capabilities of the methods described here. There exist a few directional sets of borehole antennas (e.g., [50]), but such systems are not generally available.

The linear approximation of the relation between RMS and conductivity will only hold valid within a certain range of applications. It depends on the interplay of antenna frequency, tracer conductivities and fracture apertures, as also seen in Figure 2b. Further, it depends on the validity of the thin-bed equations, which will hold true only to a certain extent in reality (e.g., [27]). However, it should be noted that while the quasi-linear dependency will likely not stay true for all applications, the relation between RMS($E_m - E_r$) and the fracture conductivity will stay monotonically increasing in most cases. Hence, the breakthrough and peak times will remain correct (with respect to the noise level), even without a strictly linear relation.

As curve tailings could not be resolved in the DRBTCs and also information on flow rates cannot be retrieved from GPR data only, established approaches for measured tracer transit time distributions such as moment analysis [51,52] cannot be applied for DRBTCs. However, if a sufficiently long time series would be available, relative moments, such as the Gini coefficient and the Peclet number, could be calculated from DRBTCs.

Because the method relies on saline tracers and thin-bed reflections, it cannot be straightforwardly extended to flow in porous media, or water flow monitoring in unsaturated media. However, also in such applications the possibility of interference phenomena could result in locally unexpected behavior of the reflected signal strength. It could thus be advisable, to investigate and employ differencing approaches also for such cases.

### 4.1.3. The Importance of Differencing in Field Data

To showcase the peculiarities of a signal strength analysis on monitoring data, we here want to present an example of amplitude reduction in the field data during the tracer injection. Figure 11a presents un-differenced traces from Experiment 2 at different times at a borehole depth of 7 m in ROI 2, which is indicated with the vertical lines. In Figure 11b the normalized, but un-differenced, monitoring RMS($E_m$) for this trace in the ROI 2 is shown in comparison with the correctly differenced DRBTC (RMS($E_m - E_r$)) of the same trace. What was described theoretically in Section 2.1 appears to be the case in this ROI: an increase in fluid conductivity led to a signal decrease in the monitoring data. This highlights that it is not reasonable to perform RMS calculations on un-differenced data. The processing of the un-differenced data here was performed completely analogue to the differenced data, with the exception of the singular value decomposition filter that was applied to the differenced data, as described in Giertzuch et al. [31].

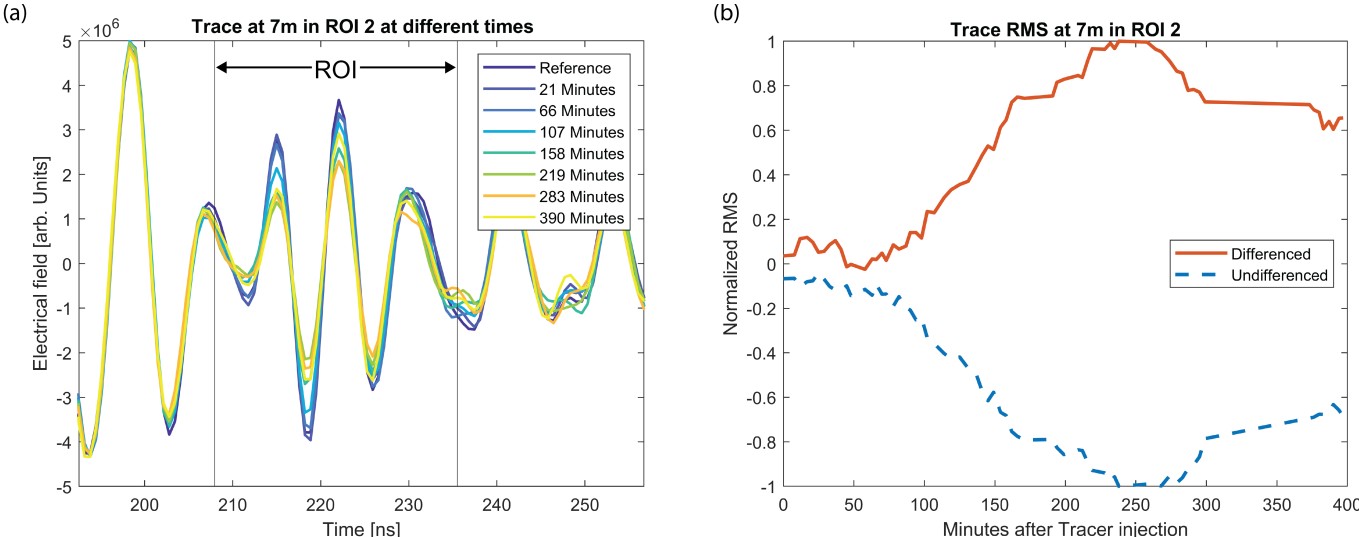

**Figure 11.** (**a**) GPR traces at different times after tracer injection. The vertical lines indicate the limits of ROI 2. (**b**) Normalized RMS($E_m$) of this monitoring trace in ROI 2 (dashed blue) in comparison with the difference RMS($E_m - E_r$) (solid orange).

### 4.2. Average Flow Path Velocity Distribution

The presented procedure to calculate the flow path averaged velocity distribution in 3D (Figure 10b) from GPR tracer data gave overall plausible results. The routing algorithm appears to be functional for the specific flow path reconstructions and returned reasonable tracer point connections, and thus flow path travel distances. However, it has to be mentioned that this performance might be limited for differently shaped flow path geometries. Furthermore, if the temporal resolution of the data set is inferior, point connections through locations that are not tracer filled are possible. These connections would have to be dealt with differently, for example, by sub-setting individual time steps into virtual time steps of higher resolution. However, we did not experience this issue for our data.

Generally, the algorithm will only yield reasonable results for the propagation fronts and for the cumulative distances, as described in Section 3.5 and shown in Figure 10a, and fails to calculate the local tracer velocities correctly. This limits the calculations to average velocities over the respective cumulative distances. A different and more sophisticated algorithm might be able to overcome this problem, but there are more limiting factors. As described for the DRBTCs, different areas in the difference profiles will pronounce the tracer at a different SNR. The tracer breakthrough is defined as the time at which the signal overcomes the background noise. This can vary between different locations. Therefore, it is possible that the tracer signal is visible above noise levels at positions associated with longer flow path distances, before it exceeds the noise level at shorter flow path distances. Furthermore, there may be no significant changes in the difference profiles over a period of time, before the tracer appears at an area with a higher SNR. Naturally, in such cases the tracer will not in fact have "jumped" between the time steps, but it was rather not visible during previous time steps. This behavior implies that local velocities (also for the tracer fronts) cannot be determined from these images. However, the average velocities presented here remain plausible, because the averaging over multiple time steps reduces this problem.

To appraise the quality of the calculated averaged tracer velocity field, we compare the findings to those of Kittilä et al. [7]. In this study, dye tracer propagation velocities between multiple observation locations were retrieved from an inversion scheme and transferred into hydraulic conductivities. The inversion in Kittilä et al. [7] operated on several tracer tests that were run during the ISC experiment, after the hydro-shearing and before the hydro-fracturing experiments at the test site were conducted [32]. The tracer GPR experiments were instead conducted after the hydro-fracturing experiments. However, as the injection interval used in the GPR experiments was not used for stimulation, we judge a comparison to be reasonable. The results of Kittilä et al. [7] are shown in Figure 12, and the part that is relevant for this study is indicated with a box. The $\log(K)$ values range between $-4$ and $-5.5$ for this area. The hydraulic conductivity, and hence the velocity around the injection point (INJ2-i4) appears to be low, and increases towards the AU tunnel. By applying the same transformation between velocity and hydraulic conductivity as described in Kittilä et al. [7], we find values for this area between $\log(K) = -3.7$ and $\log(K) = -5.1$, therefore slightly higher compared to Kittilä et al. [7]. This can be expected, as our calculation is based on the breakthrough times, whereas Kittilä et al. [7] used the peak arrival times for their calculations. Additionally, the inversion returned local velocities, but our procedure yielded averaged flow path velocities which are thus not directly comparable. However, there is a general agreement between both approaches.

In a planar fracture, flow is radial and the tracer propagation velocity decreases with distance from the injection point. However, this cannot be observed in our reconstruction, and we can therefore assume that the tracer flow appears not to be radial, but instead is dominated by preferential flow paths and flow channeling, as can be expected at our test site. It was reported that the tunnel acts as a drainage in this hydraulic system [44], and thus increasing velocities towards the tunnel would be expected. In our reconstruction this increase is visible, but close to the AU tunnel the velocities decrease again. The tunnel

outflow is located in between the S3 shear zones. We can speculate from the results presented here that the tracer velocity appears to decrease, when it leaves the S3.2 shear zone and propagates through one or multiple feature extension fractures, which then strike the AU tunnel. One reason for this velocity decrease could then be an increased fracture aperture of this fracture in comparison with the intra-S3.2 flow or an increased fracture density in this area. While around the injection borehole the fracture density is highest within the two S3 shear zones, in a borehole approximately on the elevation of the AU tunnel, a high fracture density between the two S3 shear zones was reported (Borehole SBH4, see description and Figure 11 in Doetsch et al. [39]). Other explanations could be effects from the excavation damage zone, or a desaturation of the rock volume close to the AU tunnel. However, both effects are not likely to extent further into the rock volume than approximately 2 m [53], and are thus assumed not to be the cause of the velocity decrease. The inversion cells towards the AU tunnel in Kittilä et al. [7] could only be constrained by the AU tunnel data, hence the refined information of decreasing tracer velocity towards the tunnel is an information on the GTS flow system that is exclusive to the GPR data analysis.

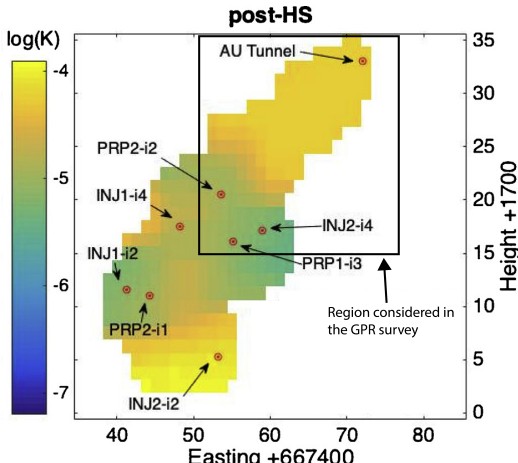

**Figure 12.** Inversion result of the hydraulic conductivity, *K*, distributions reconstructed using the peak arrival times of dye tracer breakthrough curves after the hydraulic shearing stimulations of the ISC experiment. Figure modified from Kittilä et al. [7].

*4.3. Implications for Geothermal Reservoir Characterization*

A particular challenge affecting essentially all research on geothermal reservoirs is limited accessibility and the limited spatial resolution of flow and transport observations in situ. Our results indicate that time-lapse borehole GPR can be a valuable contribution to overcome this challenge, as it allows to acquire hydrological information for otherwise inaccessible locations. Both of the methods presented provide high spatial resolution and expand the possibilities for monitoring locations significantly in comparison to local deployment of borehole sensors. However, the GPR approach is inferior in terms of SNR and data uncertainties, and can only provide relative DRBTCs. Therefore, traditional tracer and hydraulic tests should be combined with the presented GPR methods. This would allow to obtain both, locally precise, and spatially well distributed information for effective reservoir characterization. By applying the GPR methods on stimulation experiments, changes in the flow field can be characterized to assess the stimulation effects and relate changes to various locations within the reservoir. Also, a reservoir monitoring application could be feasible to determine clogging of geothermal systems.

Here, the methods were applied on an intermediate (decameter) scale experiment. Such experiments are performed in order to bridge the knowledge gap on non-linear scaling effects between between the laboratory and the full scale for geothermal applications [33,54]. However, GPR systems exist over wide frequency ranges, which makes it a particularly suitable candidate to bridge scales. In the GHz range, small scale laboratory

experiments could be performed and compared to larger scale experiments, as described in this contribution. Shakas et al. [55] have already shown that by employing 100 MHz borehole GPR stimulation induced permeability changes can be detected in a hecto-meter scale experiment. With lower frequencies, and appropriate antennas, also full scale applications could be attempted. The presented methods could then provide an effective reservoir monitoring tool, and a valuable feedback loop for stimulations within narrow target parameters could be developed and established for engineered geothermal systems.

## 5. Conclusions

In this contribution, we have demonstrated two approaches towards a quantitative analysis of time-lapse reflection GPR data for saline tracer experiments in fractured rock. We found that the relation between GPR difference reflection amplitudes, expressed in form of the RMS on difference data, and the electrical conductivity of fluids in fractures is quasi-linear for sufficiently small apertures. In contrast, the signal strength of undifferenced GPR data does not necessarily describe such a dependency, due to interference phenomena. Therefore, data differencing is recommended despite the high processing effort, especially for multi-reflector surroundings and borehole GPR. Applying the RMS measure on localized regions of interest (ROI) in difference reflection GPR data, allows for the calculation of relative tracer breakthrough curves (DRBTC) at theoretically arbitrary locations due to the described linearity.

Combining time-lapse difference reflection GPR data from two boreholes allows locating electrically conductive tracers in space and time. Here, we have developed a routing algorithm that allows to derive apparent flow path trajectories and distances from such data. This facilitates the computation of averaged tracer velocities along these flow paths, which are a better estimate than the assumption of direct connections between injection and monitoring locations.

Both approaches were tested on field data. The DRBTC calculations yielded plausible results, when compared to data from conductivity sensors in the experiment volume. Due to inherent ambiguities of borehole GPR and the limited signal-to-noise ratio of the field data, a judicious selection of the ROIs was required. Still, we could determine DRBTCs for regions in the experiment volumes that were previously inaccessible and demonstrated the capability of resolving and analyzing separate flow paths within the fracture network. Moreover, the field data confirmed the necessity to perform such calculations on difference data.

The calculated average flow path velocity distribution resulted in plausible flow path distances and the resulting velocity estimates are better approximations than direct connection assumptions. A conversion of flow path averaged velocities to local velocities is not straightforward, and there are considerable uncertainties inherent in the 4D reconstruction and the routing algorithm. However, due to the superior spatial sampling, when compared to discrete tracer monitoring locations, we judge our method to be helpful for the characterization of flow systems. For sufficiently reliable point connections and averaged flow path velocities it could be possible to retrieve the local velocities, by applying inversion techniques to such data in the future. In such an approach, it would be also possible to consider further constraints, such as fracture geometries.

In summary, GPR tracer tests with the application of the DRBTC concept and flow path velocity field calculations allowed to retrieve quantitative hydrological information for otherwise inaccessible regions. They can thus help to overcome the challenge of limited reservoir access, because they only rely on boreholes for GPR deployment, and are capable of retrieving information with high spatial resolution that is independent of possible monitoring locations. As GPR systems are available for a wide range of frequencies, applications on smaller and larger scales than the experiments described here are also reasonable. The presented methods could be a powerful add-on to conventional tracer tests, particularly for characterizing fracture networks in geothermal reservoirs.

**Author Contributions:** Conceptualization, P.-L.G., A.S., and J.D.; methodology, P.-L.G., A.S., and J.D.; software, P.-L.G.; validation, P.-L.G. and A.S.; formal analysis, P.-L.G., A.S., J.D., and H.M.; investigation, P.-L.G., J.D., B.B., and M.J.; data curation, P.-L.G.; writing—original draft preparation, P.-L.G.; writing—review and editing, P.-L.G., A.S., J.D., B.B., M.J., and H.M.; visualization, P.-L.G.; supervision, J.D. and H.M.; project administration, J.D. and H.M.; funding acquisition, J.D. All authors have read and agreed to the published version of the manuscript.

**Funding:** Funding for the ISC project was provided by the ETH Foundation with grants from Shell, EWZ, and by the Swiss Federal Office of Energy through a P&D grant. Peter-Lasse Giertzuch is supported by SNF grant 200021 169894.

**Data Availability Statement:** The data are available under: https://doi.org/10.3929/ethz-b-000456 232 (accessed on 11 May 2021).

**Acknowledgments:** The ISC is a project of the Deep Underground Laboratory at ETH Zurich, established by the Swiss Competence Center for Energy Research-Supply of Electricity (SCCER-SoE) with the support of the Swiss Commission for Technology and Innovation (CTI). The Grimsel Test Site is operated by Nagra, the National Cooperative for the Disposal of Radioactive Waste. We are indebted to Nagra for hosting the ISC experiment in their GTS facility and to the Nagra technical staff for onsite support.

**Conflicts of Interest:** The authors declare no conflict of interest.

## Abbreviations

The following abbreviations are used in this manuscript:

| | |
|---|---|
| $\varepsilon$ | dielectric permittivity |
| $\sigma$ | electrical conductivity |
| **E** | electric field |
| $t$ | time |
| BTC | breakthrough curve |
| DRBTC | difference radar breakthrough curve |
| GEO | geophysical monitoring borehole |
| GPR | ground penetrating radar |
| GTS | Grimsel Test Site |
| INJ | injection borehole |
| ISC | In-situ Stimulation and Circulation |
| NAGRA | National Cooperative for the Disposal of Radioactive Waste |
| PRP | pressure monitoring borehole |
| RMS | root mean square |
| ROI | region of interest |
| SBH | stress monitoring borehole |
| SNR | signal to noise ratio |

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
