# Peer review of "Computing Localized Breakthrough Curves and Velocities of Saline Tracer from Ground Penetrating Radar Monitoring Experiments in Fractured Rock"

_energies, doi:10.3390/en14102949_

Round 1
Reviewer 1 Report
The following correction should be done before further processing;
1- In the introduction section, you should discuss and compare the novelty of your work with other literature and say why your work is novel and highlight the novelty in comparison with previous works. The following references are recommended to cite too.'
- Neutrally buoyant tracers in hydrogeophysics: Field demonstration in fractured rock
- Time-lapse reflection and transmission borehole GPR for saline tracer monitoring in fractured rock
- Integrated production logging tools approach for convenient experimental individual layer permeability measurements in a multi-layered fractured reservoir
- Parametric study of polymer-nanoparticles-assisted injectivity performance for axisymmetric two-phase flow in EOR processes
- Transient hydraulic tomography approach to characterize main flowpaths and their connectivity in fractured media
2- Please don't use abbreviations in the title and abstract for the first time
3- Please add the nomenclature section to the paper
4- Could you please explain the limitations of the model in more detail?
5- Improve your conclusion section
Reviewer 2 Report
One of the central challenges in groundwater hydrology is to accurately characterize fractured rock heterogeneity. Flow and transport in fractured rock can be very unpredictable, partly because of the highly anisotropic, multicomponent, and multi-scale behavior of fractured rocks. Obtaining sufficient information to characterize fractured rock heterogeneity through hydrological tests alone is difficult, partly because hydrological data is often spatially sparse as it is measured along boreholes. For example, artificial tracer tests involve the injection of a volume of tracer in the fractured system and its (partial) recovery, either at another location during a dipole test or at the injection location during a push-pull test. The measured tracer concentration during the recovery phase is called the breakthrough curve (BTC). Heterogeneities in the geometrical properties of fractures often manifest as low concentrations of tracer arriving at later times and a resulting BTC that is skewed. Flow and transport modeling can be used to fit the shape of the BTC and to implicitly characterize heterogeneity. Nevertheless, tracer data lack direct information, at least, about (1) the contact area between the tracer and the fracture walls (i.e., number and size of fractures involved), (2) the dynamics of the tracer away from the borehole location and (3) ambient flow in the system. Accordingly, hydrogeophysical investigations have met success since they can partially provide such complementary information through geophysical techniques.
I personally consider that the work undertaken by the authors is valuable, the results obtained are clearly presented, and the conclusions sufficient and to the point.
Due to the complexity of the phenomenon analyzed in the paper, the authors should take into account many more works already done by increasing both bibliographic references and references to them, but especially to try to include several methods already used successfully in the past.
